# Impact of type of dialyzable beta-blockers on subsequent risk of mortality in patients receiving dialysis: A systematic review and meta-analysis

Tzu-Hsuan Yeh[1], Kuan-Chieh Tu[1], Kuo-Chuan Hung[2], Min-Hsiang Chuang[1], Jui-Yi Chen[3,4]*

1 Department of Internal Medicine, Chi Mei Medical Center, Tainan, Taiwan, 2 Department of Anesthesiology, Chi Mei Medical center, Tainan, Taiwan, 3 Division of Nephrology, Department of Internal Medicine, Chi Mei Medical Center, Tainan, Taiwan, 4 Department of Health and Nutrition, Chia Nan University of Pharmacy and Science, Tainan, Taiwan

* kwuilus0101@gmail.com

**Data Availability Statement:** All relevant data are within the manuscript and its Supporting Information files.

## Abstract

### Background

Beta-blockers has been reported to improve all-cause mortality and cardiovascular mortality in patients receiving dialysis, but type of beta-blockers (i.e., high vs. low dialyzable) on patient outcomes remains unknown. This study aimed at assessing the outcomes of patients receiving dialyzable beta-blockers (DBBs) compared to those receiving non-dialyzable beta-blockers (NDBBs).

### Methods

We searched the databases including PubMed, Embase, Cochrane, and ClinicalTrials.gov until 28 February 2022 to identify articles investigating the impact of DBBs/NDBBs among patients with renal failure receiving hemodialysis/peritoneal dialysis (HD/PD). The primary outcome was risks of all-cause mortality, while the secondary outcomes included risk of overall major adverse cardiac event (MACE), acute myocardial infarction (AMI) and heart failure (HF). We rated the certainty of evidence (COE) by Cochrane methods and the GRADE approach.

### Results

Analysis of four observational studies including 75,193 individuals undergoing dialysis in hospital and community settings after a follow-up from 180 days to six years showed an overall all-cause mortality rate of 11.56% (DBBs and NDBBs: 12.32% and 10.7%, respectively) without significant differences in risks of mortality between the two groups [random effect, aHR 0.91 (95% CI, 0.81–1.02), $p = 0.11$], overall MACE [OR 1.03 (95% CI, 0.78–1.38), $p = 0.82$], and AMI [OR 1.02 (95% CI, 0.94–1.1), $p = 0.66$]. Nevertheless, the pooled odds ratio of HF among patients receiving DBBs was lower than those receiving NDBB

**Funding:** This study was supported by Chi-Mei Medical Center (CMFHR10973) The funders had no role in study design, decision to publish, data collection and analysis, or preparation of the manuscript.

**Competing interests:** The authors declare that they have no competing interests.

[random effect, OR 0.87 (95% CI, 0.82–0.93), *p*<0.001]. The COE was considered low for overall MACE, AMI and HF, while it was deemed moderate for all-cause mortality.

## Conclusions

The use of dialyzable and non-dialyzable beta-blockers had no impact on the risk of all-cause mortality, overall MACE, and AMI among dialysis patients. However, DBBs were associated with significant reduction in risk of HF compared with NDBBs. The limited number of available studies warranted further large-scale clinical investigations to support our findings.

## Introduction

Adrenergic beta receptors play an important role in maintaining the normal physiological functions particularly in the nervous and cardiovascular (CV) systems [1]. Beta-blockers (BBs), which are competitive antagonists that block the receptor sites for endogenous catecholamines (i.e., epinephrine and norepinephrine) [2], are a class of medications commonly used in the treatment of cardiovascular diseases including high blood pressure, coronary artery disease (CAD), or heart failure (HF) with a reduced ejection fraction (rEF) [3–8]. Additionally, a previous placebo-controlled clinical trial showed the beneficial effect of a BB on reducing the risks of all-cause and CV mortality in patients with dilated cardiomyopathy undergoing dialysis compared with non-users [9]. Consistently, a meta-analysis of three randomized controlled trials (RCTs) demonstrated an association of the use of BBs with reduced risks of all-cause mortality, CV mortality, CV events, and hospitalizations for dialysis patients [1]. However, although the molecular weight, protein binding capacity, and volume distribution of BBs are known to affect their filtering ability from the artificial kidney membrane, the impacts of dialyzability of BBs was not investigated in those studies [10]. Dialyzable BBs (DBBs) consist of atenolol, acebutolol, metoprolol, bisoprolol, and nadolol, while carvedilol, labetalol, and propranolol belong to the category of non-dialyzable BBs (NDBBs) [11, 12]. However, there was no definite threshold for differentiating between DBBs and NDBBs. Although a previous clinical study demonstrated a beneficial impact of NDBBs on all-cause mortality due to their relatively high drug concentrations in dialysis patients [12], other studies concluded that dialysis patients receiving NDBBs were associated with a higher risk of mortality because of an elevated risk of intradialytic hypotension [13, 14]. As a result, the influence of the DBBs/NDBBs in patients undergoing dialysis remains controversial. Therefore, we conducted a systematic review and meta-analysis in an attempt to provide comprehensive evidence regarding the impact of DBBs/NDBBs on the prognostic outcomes of dialysis patients. In addition, we hypothesized that the use of DBBs was associated with a decreased risk of all-cause mortality. Hence, the primary outcome of the current study was the all-cause mortality and the secondary outcomes were the risks of overall major adverse cardiac events (MACE), acute myocardial infarction (AMI), and heart failure (HF).

## Materials and methods

This meta-analysis followed the guidelines of the Preferred Reporting Items for Systematic Reviews and Meta-Analyses (PRISMA) statement [15] and Cochrane methods [16]. The systematic review protocol was prespecified and registered in PROSPERO [CRD42022307769].

## Strategy of literature search

We searched for articles that evaluated the association of the use of DBBs and NDBBs with mortality in patients undergoing dialysis in four databases including PubMed, Embase, Cochrane, and ClinicalTrials.gov from the inception dates to 28 February 2022 without restriction on language or geographic locations. The articles were selected by manual screening. The following search terms were used: "Adrenergic beta-Antagonists," "beta-Receptor Blockaders," "beta Blockers, Adrenergic," "Renal Dialysis," "hemodialysis," "Mortality," and "Death." Two investigators (TH Yeh, KC Tu) searched the published observational studies and checked all articles separately to prevent bias. Full-text papers were selected for quality assessment and data syntheses. We contacted the authors of the included articles with missing data to acquire additional information.

## Inclusion and exclusion criteria

Studies were considered eligible if the following criteria were met: (a) Population: adults ($\geq$ 18 years old) with end-stage renal disease (ESRD) on maintenance dialysis; (b) Exposure group: patients receiving DBBs; (c) Control group: patients receiving NDBBs (d) Outcome: risk of all-cause mortality, incidence of overall MACE, AMI, or HF. The exclusion criteria were (1) studies recruiting individuals less than 18 years old; (2) those in which information regarding BBs or outcomes was unavailable; (3) publications such as reviews, letters, conference abstracts, case reports, or studies other than original investigations. Two authors independently examined the titles and abstracts of the included articles to determine their eligibility for the final analysis. All disagreements were resolved through discussion.

## Data extraction

For each eligible study, general information (first author, year of publication, year of study, study name, study design, sample size), baseline demographic and clinical characteristics of the participants (population, age, percentage of male, country, kidney replacement therapy (KRT) modality, duration of dialysis, comorbidities, dialyzability of BBs, follow-up duration), interventions/exposure (BB treatment), and outcome data [e.g., MACE, AMI, HF and all-cause mortality] were extracted. Information about all-cause mortality was acquired from the healthcare database of each country, which specified the date and cause of death. The definition of MACE was based on that of each included study.

## Quality assessment

We used the Newcastle-Ottawa Scale (NOS) to assess the risk of bias of the included studies. The following three domains, which consisted of eight items, were evaluated: the representativeness of exposed cohort (one point), selection of non-exposed cohort (one point), ascertainment of exposure (one point), outcome of the interest not present at the start of the study (one point), comparability of cohorts (two point), assessment of outcome (one point), follow-up duration (one point), and adequacy of follow-up of cohorts (one point). Studies having 7–9, 4–6, and 0–3 points were considered to be of high, moderate, and low quality, respectively. We rated the certainty of evidence (COE) according to the Cochrane methods and the GRADE approach.

## Subgroup analysis and sensitivity analysis

Subgroup and sensitivity analyses were conducted for assessing heterogeneity among the included studies. Subgroup analyses were performed focusing on the effects of age ($<$ 65 vs.

≥65), proportion of CAD individuals (< 50% vs. ≥ 50%), follow-up duration (≦one year vs. >one year) and scores of Newcastle-Ottawa Scale Quality Assessment (NOS score 8 vs. 9) on the mortality rates. In addition, we conducted a sensitivity analysis to evaluate the impact of each study on the overall estimate by omitting one article at a time.

## Data synthesis and statistical analysis

Analyses were performed to compare the outcomes of exposure between patients receiving DBBs and those being administered NDBBs. The odds ratio (OR) and 95% confidence interval (CI) were extracted from the included studies. In addition, the hazard ratio (HR) of mortality was also analyzed. For studies that provided information on HR, the data were used for analysis. Random effects model was used to analyze outcomes, namely the risks of all-cause mortality, overall MACE, AMI, and HF between the two groups. The effect size is expressed as the pooled OR and 95% CI. Egger's test and Begg's test were used to examine potential publication bias. Between-trial heterogeneity was determined by $I^2$ tests and values >50% were regarded as considerable heterogeneity [16]. A probability value ($p$) less than 0.05 was defined as statistically significant. We used Comprehensive Meta-Analysis (Version 3.3.070, November 20, 2014) for all statistical analyses.

## Results

### Outcomes of literature search and included patients

In this study, 563 studies were identified through searching the PubMed, Embase, Cochrane, and ClinicalTrials.gov database and 320 records were retrieved through registry or other sources. After removal of 345 duplicates and 531 articles following screening of their titles and abstracts, seven studies were assessed through full-text review that excluded three articles for lacking available data. Therefore, a total of four articles were deemed eligible for this meta-analysis (Fig 1).

### Study characteristics

All four included papers were retrospective observational studies focusing on the comparison between the use of DBBs and NDBBs in patients on maintenance dialysis. The baseline

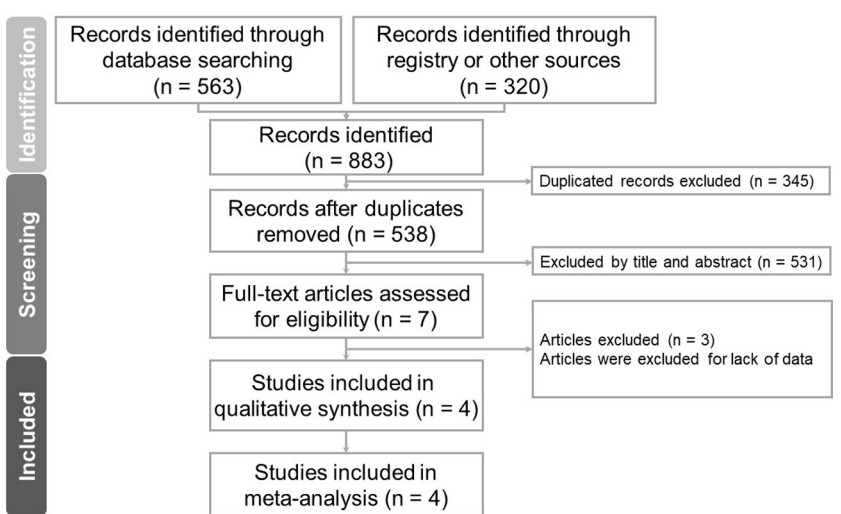

**Fig 1. Flow chart for study selection.**

characteristics of the included studies are shown in Tables 1 and 2. The four included studies were published between 2015 to 2020 and enrolled a total of 75,193 dialysis patients. Among the included articles, the number of patients ranged from 4,938 to 36,603 with age between 56.64 and 75.65 years. The classification of DBBs and NDBBs was based on the pharmacokinetics of beta-blockers [17–22] in two of the included studies [10, 12], while the classification of the other two studies was based on the cardio-selectivity of beta-blockers [13, 23]. Previous studies have shown that cardio-selective and non-cardio-selective beta-blockers are pharmacologically equivalent to DBBs and NDBBs, respectively [11, 24]. Among the included patients, 40,280 and 34,913 were exposed to DBBs and NDBBs, respectively. Regarding KRT modality, three of the studies recruited participants undergoing only hemodialysis [10, 12, 13], while the other study enrolled patients receiving concomitant hemodialysis and peritoneal dialysis [23]. The follow-up duration ranged from 180 days to two years in three studies, while one study followed their patients till the time of the first outcome event (i.e., death or cardiovascular event) [23]. The four studies were from the USA [13, 23], Canada [12], and Taiwan [10].

## Quality of enrolled studies and publication bias

The quality of the included studies was deemed high according to the NOS tool (median score, 9; range, 8 to 9) (S1 Table). Of the four included studies, one reported the shortest follow-up period (i.e., 180 days) [12] was given only eight points. There was no significant publication bias in all four studies according to Begg's rank correction and Egger's weighted regression analyses ($p>0.05$).

## Primary outcome: Association of dialyzability of beta-blockers with risk of all-cause mortality

Analysis of the all-cause mortality on 75,193 patients, the main outcome of interest, from all four included studies showed a pooled incidence of 11.56% (i.e., 8,694 patients). The pooled

**Table 1. Summary of the baseline characteristics of the included studies (n = 4).**

| Author | Data sources/Country | Study period/ Follow-up (duration) | Mean age (years) | Male (%) | CAD (%) | Patients (n) Total (DBBs vs. NDBBs) | DBBs | NDBBs | KRT Modality |
|---|---|---|---|---|---|---|---|---|---|
| Weir (2015) | healthcare databases/Canada | 2002-2011/180 days | 75.65 | 49.1 | 62.6 | 6498 (3294 vs. 3294) | Atenolol, Acebutolol, Metoprolol | Bisoprolol, Propranolol | HD |
| Shireman (2016) | USRDS and Medicaid (Centers for Medicare & Medicaid Services or CMS)/ USA | 2000-2005/6 years | 59.9 | 43.36 | 26.4 | 4938 (3781 vs. 1157) | Atenolol, Metoprolol | Carvedilol, Labetalol | HD/PD |
| Assimon (2018) | USRDS data/USA | 2007-2012/1 year | 59.6 | 53.3 | 29.8 | 4938 (3781 vs. 1157) | Metoprolol | Carvedilol | HD |
| Wu (2020) | Taiwan NHIRD/Taiwan | 2004-2011/2 years | 56.64 | 48.9 | 32.7 | 36603 (15699 vs. 20904) | Atenolol, Acebutolol, Metoprolol, Bisoprolol | Betaxolol, Carvedilol, Popranolol | HD |

Abbreviations:

BBs, beta-blockers; CAD, coronary artery disease; CMS, The Centers for Medicare and Medicaid Services; HD, hemodialysis; DBBs, dialyzable beta-blockers; KRT, kidney replacement therapy; NDBBs, non-dialyzable beta-blockers; NHIRD, National Health Insurance Research Database; PD, peritoneal dialysis; USA, The United States of America; USRDS, The United States Renal Data System

**Table 2. Summary of included studies for outcome evaluation.**

| Author | Time point of BBs given | Primary outcome | Secondary outcome | All-cause mortality | | CVMM**/overall MACE* | |
|---|---|---|---|---|---|---|---|
| | | | | DBBs vs NDBBs (%) | HR (95% CI) | DBBs/ NDBBs (%) | HR (95% CI) |
| Weir (2015) | After HD at least 120 days | Mortality | CV mortality, CV disease (MI, HF) | 5.53 vs. 4.1 | 1.3‡ (1.1–1.7) | 8.17/6.83 | 1.2 (1.0–1.5)¶ ‡ |
| Shireman (2016) | Days 91–180 after initiating dialysis | ACM/CVMM | NR | 32.95 vs. 32.76 | 0.84 (0.72–0.97) | 46.55/46.55 | 0.86 (0.75–0.99) |
| Assimon (2018) | 180-day baseline period free of any oral BBs use | All-cause and CV mortality | All-cause and CV hospitalizations | 15.26 vs.17 | 0.93 (0.86–0.98) | NR | NR |
| Wu (2020) | 90-day baseline period free of any oral BBs use | All-cause mortality and MACE* | NR | 5.5 vs. 7.62 | 0.82 (0.75–0.88) | 10.7/22.39 | 0.89 (0.84–0.93) |

Abbreviations:

ACM, All-cause mortality; aHR, adjusted hazard ratio; BBs, Beta-blockers; CHF, congestive heart failure; CI, Confidence interval; CV, cardiovascular; CVA, cerebrovascular accident; CVMM, Cardiovascular morbidity and mortality; HD, hemodialysis; DBBs, dialyzable beta-blockers; HF, heart failure; IHD, ischemic heart disease; NDBBs, non-dialyzable beta-blockers; MACE, major adverse cardiac event; MI, myocardial infarction; NR, Not report; PVD, peripheral vascular disease

*MACE: hospital admission with a primary diagnosis of acute myocardial infarction, heart failure or ischemic stroke

**CVMM: defined as inpatient hospitalization for myocardial infarction, ischemic heart disease, revascularization, congestive heart failure, cerebrovascular accident, or peripheral vascular disease. Cardiovascular-related mortality caused by myocardial infarction, atherosclerotic heart disease, cardiomyopathy, cardiac arrhythmia, cardiac arrest, cerebrovascular accidents.

¶ relative risk(RR)

‡ adjust HR

incidence of all-cause mortality in those receiving DBBs and NDBBs was 12.32% (4962 of 40280) and 10.7% (3732 of 34913), respectively. Examination of the pooled results revealed no difference in risk of mortality between patients receiving DBBs and those being given NDBBs [Fig 2A, random effects, aHR 0.91 (95% CI, 0.81–1.02), $p$ = 0.11].

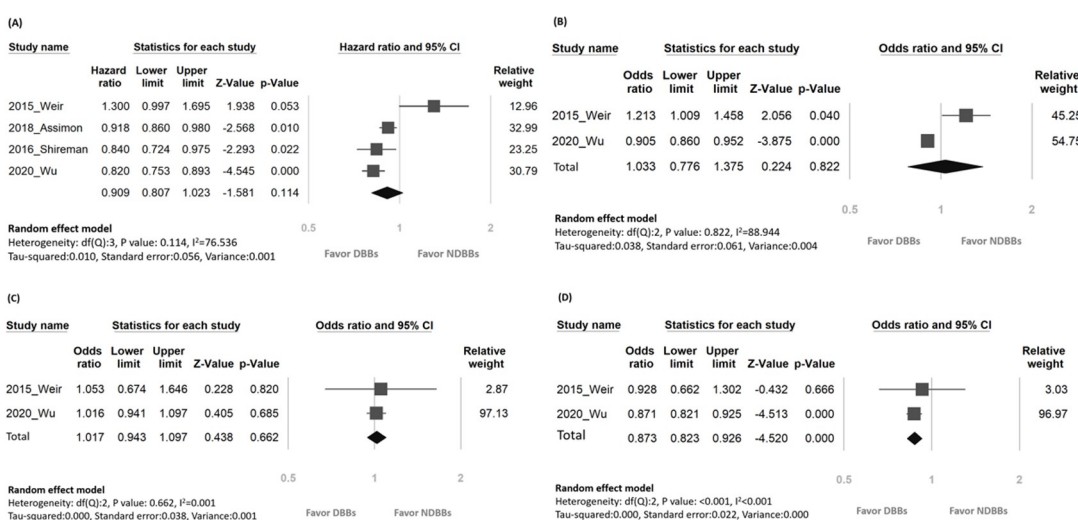

**Fig 2.** Forest plots comparing the risks of (A) all-cause mortality (aHR), (B) overall major adverse cardiac events (MACE), (C) acute myocardial infarction (AMI), (D) Heart failure (HF) between patients receiving dialyzable beta-blockers and those with non-dialyzable beta-blockers. aHR, adjusted hazard ratio; DBBs, dialyzable beta-blockers; NDBBs, non-dialyzable beta-blockers; CI, confidence interval.

## Secondary outcome: Association of dialyzability of beta-blockers with risk of overall MACE

The pooled incidence of overall MACE from two available studies including 43,191 patients [10, 12] was 19.5% (i.e., 8,424 patients). There was a slight variation in the definition of overall MACE between the two studies; while it involved cardiovascular death, myocardial infarction, and heart failure in one of the studies [12], the other referred it to acute coronary syndrome, ischemic stroke, and heart failure [10]. Based on the two studies, the pooled ratios of overall MACE associated with DBBs and NDBBs were 18.53% (3519 of 18993) and 20.27% (4905 of 24198), respectively. The pooled odds ratio of overall MACE was non-significantly higher in the DBB group than that in the NDBB group [Fig 2B, random effect, OR = 1.03 (95% CI, 0.78–1.38), $p$ = 0.82]. Similarly, the risk of AMI was non-significantly higher in the former than that in the latter [Fig 2C, random effect, OR = 1.02 (95% CI, 0.94–1.10), $p$ = 0.66]. However, the pooled odds ratio of HF was significantly lower in patients receiving DBBs compared to those in the NDBB group [Fig 2D, random effect, OR = 0.87 (95% CI, 0.82–0.93), $p$ < 0.001].

## Subgroup analysis

In studies that recruited a relatively small proportion of patients diagnosed with CAD (i.e., <50%) [10, 13, 23], the use of DBBs was associated with a non-significantly lower risk of mortality compared to that with NDBBs [Fig 3A, random effect, OR = 0.85 (95% CI, 0.71–1.02), $p$ = 0.09]. In contrast, for studies enrolling a high proportion of patients with CAD (i.e., ≥ 50%) [12], the use of DBBs was related to a non-significantly higher risk of mortality compared to that with NDBBs [Fig 3A, random effect, OR = 1.369 (95% CI 0.934–2.005), $p$ = 0.107]. Regarding the impact of age on mortality, among studies that included relatively young patients (i.e., age < 65) [10, 13, 23], the use of DBBs was associated with a non-significant reduction in mortality risk compared to NDBBs [Fig 3B, random effect, OR = 0.85 (95% CI, 0.71–1.02), $p$ = 0.09]. In contrast, focusing on studies that recruited older individuals on dialysis [12], there was a non-significant increase in mortality risk among those receiving DBBs compared with those taking NDBBs [Fig 3B, random effect, OR = 1.369 (95% CI, 0.934–

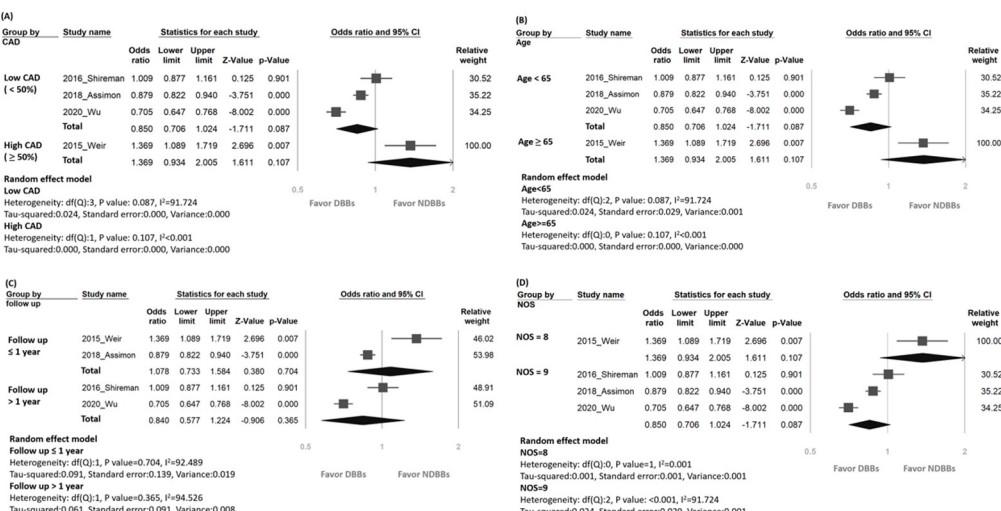

**Fig 3.** Forest plots of subgroup analysis of the effects of (A) low (<50%) vs. high (≥50%) proportion of patients with coronary artery disease (CAD), (B) Age <65 vs. ≥65 years, (C) follow-up duration ≦1 vs. >1 year, (D) Newcastle-Ottawa Scale Quality Assessment (NOS) score 8 vs. 9 on mortality risk of patients on dialysis receiving beta-blockers. DBBs, dialyzable beta-blockers; NDBBs, non-dialyzable beta-blockers; CI, confidence interval.

2.005), *p* = 0.107]. Taken together, despite the lack of statistical significance, there was a trend that relatively young (i.e, age less than 65) dialysis patients without the diagnosis of CAD had a relatively lower mortality risk if they received DBBs compared to those using NDBBs. With respect to the effect of follow-up duration on mortality, analysis of the studies with follow-up durations shorter than a year [12, 13] showed a non-significantly higher mortality risk associated with DBBs than that with NDBBs [Fig 3C, random effect, OR = 1.08 (95% CI, 0.73–1.58), *p* = 0.70]. On the other hand, focusing on studies with follow-up durations over one year [10, 23] demonstrated a non-significantly lower mortality risk in those using DBBs than those in the NDBB group [Fig 3C, random effect, OR = 0.84 (95% CI, 0.58–1.22), *p* = 0.37]. In respect of the impact of the quality of the included studies, analysis of the study with a NOS of eight showed a higher but non-significant increase in mortality risk in the DBB group compared to that in the NDBB group [Fig 3D, random effect, OR = 1.37 (95% CI, 0.93–2.0), *p* = 0.12]. In contrast, investigating the other three studies reporting a NOS score of nine demonstrated a non-significant reduction in mortality risk among those receiving DBBs compared with those taking NDBBs [Fig 3D, random effect, OR = 0.85 (95% CI, 0.71–1.02), *p* = 0.09].

## Sensitivity analysis

Exclusion of the study that recruited patients undergoing peritoneal dialysis [23] and another study that categorized bisoprolol as a NDBB [12] demonstrated no significant impact on the overall results (S2 Table).

## Meta-regression analysis

The quantitative measures of CAD and HF were not associated with all-cause mortality. (CAD, Z = 1.65, *p* = 0.10; HF, Z = 1.28, *p* = 0.20, S3 Table)

## Assessment of evidence quality and summary of findings

According to the GRADE system, the quality of evidence for outcomes was judged to be low (i.e., risk of overall MACE, AMI and HF) to moderate (i.e., all-cause mortality) (S1 Fig).

## Discussion

Although the use of beta-blockers is common among patients on maintenance dialysis [25], the clinical impact of their dialyzability has not been adequately addressed. The current study, which comprehensively investigated the dialyzability of beta-blockers on subsequent risks of mortality and overall MACE in patients receiving dialysis, had several clinical implications. Our analysis of four observational studies suggested that dialyzability of beta-blockers was not associated with the risks of mortality, overall MACE, and AMI among dialysis patients. On the other hand, DBBs were related to a significant reduction in the risk of HF compared with NDBBs. Moreover, there was a trend that dialysis patients diagnosed with CAD and those with age over 65 years had a relatively lower mortality risk if they received NDBBs compared with DBBs. In contrast, studies with a follow-up duration beyond one year showed a relatively lower mortality risk among those receiving DBBs compared to those using NDBBs.

Patients diagnosed with ESRD are at a high risk of cardiovascular diseases because of renal failure-related sympathetic nervous system activation [23, 25]. For instance, excessive sympathetic drive is associated with left ventricular hypertrophy known to contribute to reduced survival rates among ESRD patients [23, 26]. Although beta-blockers are commonly prescribed for this patient population to alleviate sympathetic tone [27], the clinical impacts of their heterogeneous pharmacological properties remain unclear. In addition to the difference in

dialyzability and cardiac sensitivity, variations in their pharmacodynamics, pharmacokinetics, and side-effects have also been reported [28]. For example, the concentration of beta-blockers could be highly variable, depending on the route of excretion (i.e., renal or hepatic) as well as dialyzability [28]. Furthermore, differences in their volume of distribution, percentage of protein binding, molecular weight, and modes of dialysis are known to be related to the efficiency of drug removal during dialysis. For instance, hypoalbuminemia has been reported to reduce the therapeutic dosage through lowering the proportion of drug protein-binding form and increasing its free concentrations [29, 30].

In terms of categories, the high cardio-selectivity and low lipophilicity of atenolol and metoprolol may explain their dialyzability, while carvedilol, labetalol, and propranolol are considered non-dialyzable because of their high lipophilicity [23]. The previous finding that cardio-selective beta-blockers could provide better therapeutic outcomes for patients with HF compared to those with low cardio-selectivity [31] suggested that dialyzability may affect the efficacy of beta-blockers in patients on dialysis. Nevertheless, the current study revealed no difference in the risk of mortality between dialysis patients receiving DBBs and those being given NDBBs. Regarding the classification of beta-blockers according to their dialyzability, there has been some controversy over bisoprolol whose dialytic clearance (44 mL/min) falls between the dialyzable agents (e.g., atenolol: 72 mL/min; metoprolol: 87 mL/min) and those that are considered non-dialyzable (e.g., carvedilol: 0.2 mL/min) [11]. While bisoprolol was considered non-dialyzable in one of our included trials [12], other more recent studies showed evidence of its removal during hemodialysis [10, 11, 13]. To address this issue, we conducted sensitivity analysis through omitting the previous study that classified bisoprolol as non-dialyzable [12] and found no significant impact on our overall outcome. Besides dialyzability, the therapeutic efficacy of certain beta-blockers could also be affected by hepatic function [28]. For instance, although most DBBs in the included studies are excreted through the kidneys, some are eliminated through the hepatic routine (e.g. acebutolol, bisoprolol). Similarly, while most NDBBs in our included studies are excreted by the liver, propranolol is eliminated by the kidneys.

Of the four included studies, one reported a lower risk of mortality with the use of NDBBs compared to that with DBBs [12]. One possible explanation may be the progressive diminution of circulating level of the latter during dialysis. On the contrary, the other three studies that demonstrated a survival advantage in the DBB group attributed their findings to an elevated risk of intradialytic hypotension associated with those receiving NDBBs [10, 11, 13]. Indeed, intradialytic hypotension has been shown to correlate with a higher risk of all-cause and CV mortality in the hemodialysis population [13, 32]. In contrast, some studies reported that carvedilol (i.e., NDBB) may offer cardiovascular advantages, especially in patients with hypertension or/and heart failure [31, 33]. Therefore, in addition to dialyzability of beta-blockers, the underlying condition of individuals undergoing dialysis could affect the prognostic impacts of these agents.

Our secondary outcome analysis revealed no significant association of the dialyzability of beta-blockers with the risks of overall MACE and AMI. Interestingly, the two studies that provided information about MACE [10, 12] presented controversial results. While one study demonstrated that NDBBs were related to a lower risk of overall MACE [12], the other reported opposite findings [10]. On the other hand, there was no significant difference in the risk of MI between the two groups in the two included studies [10, 12]. The European Society of Cardiology (ESC) guidelines recommend the use of beta-blockers in those with recent MI because of evidence supporting a significant reduction in mortality and cardiovascular events in this patient population. Moreover, beta-1 selective blockers (i.e. bisoprolol, metoprolol, or nebivolol) may be preferred due to less side effects (e.g., bronchospasm) [34]. Regarding HF, our

results showed a significant association between the use of DBBs and a reduction in risk of HF in patients on dialysis. According to the American Heart Association (AHA)/American College of Cardiology/Heart Failure Society of America 2022 guidelines for the management of heart failure, blood pressure control is essential for reducing the risk of HF [35]. Some studies revealed that beta-1 selective blockers have slightly greater antihypertensive effect than that of non-cardio-selective blockers [36]. Therefore, the fact that the DBBs used in our included studies all belong to cardio-selective beta-blockers may at least partly explain our finding that the participants receiving DBBs had a lower risk of HF than those with NDBBs. However, this conclusion was drawn from only two studies, one [10] of which contributed to 97% of the relative weight while the finding of the other [12] was questionable because the authors classified bisoprolol into the DBB category.

Albeit statistically non-significant, or subgroup analysis showed a trend that studies that recruited dialysis patients with a high proportion of CAD ($\geq$ 50%) or those who aged more than 65 years had a lower mortality risk if the patients received NDBBs compared with those being given DBBs. Taking into account the benefits of beta-blockers in the treatment of CAD including a reduction in myocardial oxygen demands via heart rate suppression, an increase in coronary blood flow through increasing the diastolic perfusion time, and protection against microvascular damage [37], we speculate that NDBBs may reach a higher serum concentration to attain a more effective protective effect compared with DBBs among patients with CAD. In the group of age less than 65 years old, we presume that the population of younger people do have a lower mortality rate. Our subgroup analysis further demonstrated that individuals on dialysis with a follow-up duration over one year had a relatively lower mortality risk if they received DBBs compared with NDBBs. The reported therapeutic advantage of cardio-selective beta-blockers (i.e., DBBs) based on the ESC and AHA guidelines [34, 35] may support our finding.

## Strengths and limitations

The current systematic review and meta-analysis is the first to comprehensively investigate the impact of beta-blocker dialyzability on mortality among dialysis patients through pooling updated results from clinical studies. Besides, we showed a significant correlation between the use of DBBs and a reduced risk of HF among individuals on dialysis. On the other hand, our finding of a lack of significant association of beta-blocker dialyzabiliy with all-cause mortality as well as the risks of overall MACE and AMI suggested that drug dialyzability may not be the only factor affecting mortality in this patient population. Nevertheless, this study had several limitations. First, the inclusion of bisoprolol, a beta-blocker of intermediate dialyzability, may bias our findings. In particular, while three studies classified bisoprolol into the DBB category, it was considered a NDBB in the other study [12]. Second, one out of our four studies included participants undergoing peritoneal dialysis [23], which may also affect our results. Third, the inclusion of relatively old patients in one of the studies [12] compared with the other three reports may be another confounder in the present study. Finally, because of the reliability of the funnel plot in detecting publication bias only when 10 or more outcomes are available [38], the current meta-analysis included only four studies so that their publication bias remains to be elucidated. Therefore, further well-designed studies are warranted to address these issues.

## Conclusions

The current meta-analysis, which is the first to investigate the effect of dialyzability of beta-blockers on mortality in dialysis patients, showed that dialyzabiliy was not associated with the risks of mortality, overall major adverse cardiovascular events, and acute myocardial infarction in this patient population. However, individuals receiving dialyzable beta-blockers were

associated with a significant reduction in the risk of heart failure compared with those using non-dialyzable blockers. In addition, there was a non-significant trend of an association between the use of dialyzable beta-blockers and a survival advantage among those without coronary artery disease and in relatively young subjects. Further large-scale randomized controlled trials are needed to verify our findings.

## Supporting information

**S1 Checklist. PRISMA checklist.**
(DOCX)

**S1 Text. Search strategies.**
(DOCX)

**S1 Table. Quality assessment of the included studies.**
(DOCX)

**S2 Table. Sensitivity analyses.**
(DOCX)

**S3 Table. Meta-regression analysis.**
(DOCX)

**S1 Fig. Quality assessment the GRADE results.**
(DOCX)

## Acknowledgments

The authors thank the staff of Nephrology division, Chi-Mei Medical center with any support for this study.

## Author Contributions

**Conceptualization:** Kuo-Chuan Hung, Jui-Yi Chen.

**Data curation:** Kuo-Chuan Hung, Min-Hsiang Chuang, Jui-Yi Chen.

**Formal analysis:** Tzu-Hsuan Yeh.

**Funding acquisition:** Tzu-Hsuan Yeh.

**Investigation:** Kuan-Chieh Tu, Min-Hsiang Chuang, Jui-Yi Chen.

**Methodology:** Kuan-Chieh Tu, Jui-Yi Chen.

**Project administration:** Kuan-Chieh Tu, Jui-Yi Chen.

**Resources:** Kuan-Chieh Tu, Jui-Yi Chen.

**Software:** Jui-Yi Chen.

**Supervision:** Jui-Yi Chen.

**Validation:** Tzu-Hsuan Yeh, Jui-Yi Chen.

**Visualization:** Tzu-Hsuan Yeh, Kuo-Chuan Hung, Jui-Yi Chen.

**Writing – original draft:** Tzu-Hsuan Yeh, Min-Hsiang Chuang.

**Writing – review & editing:** Tzu-Hsuan Yeh, Kuo-Chuan Hung.

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
