## [Decision Letter · Decision Letter 0]

24 Oct 2022

PONE-D-22-24265Impact of type of dialyzable Beta-blockers on subsequent risk of mortality in patients receiving dialysis: A Systematic Review and Meta-AnalysisPLOS ONE

Dear Dr. Chen,

Thank you for submitting your manuscript to PLOS ONE. After careful consideration, we feel that it has merit but does not fully meet PLOS ONE’s publication criteria as it currently stands. The referees found your work of potential interest, but they also noted several issues that deserve attention (see their reports below). We invite you to submit a revised version of the manuscript that addresses all issues thoroughly and unequivocally. Please note that both referees had several methodological concerns that should be dealt with with the utmost care. 

We look forward to receiving your revised manuscript.

Kind regards,

Gianpaolo Reboldi, MD, MSc, PhD

Academic Editor

PLOS ONE

Journal Requirements:

"This study was supported by Chi-Mei Medical Center (CMFHR10973)

The funders had no role in study design, decision to publish, data collection and analysis, or preparation of the manuscript."

"This study was supported by Chi-Mei Medical Center (CMFHR10973)

The funders had no role in study design, decision to publish, data collection and analysis, or preparation of the manuscript."

4. We note that this manuscript is a systematic review or meta-analysis; our author guidelines therefore require that you use PRISMA guidance to help improve reporting quality of this type of study. Please upload copies of the completed PRISMA checklist as Supporting Information with a file name “PRISMA checklist”.

Reviewers' comments:

Reviewer's Responses to Questions

**Comments to the Author**

1. Is the manuscript technically sound, and do the data support the conclusions?

Reviewer #1: Partly

Reviewer #2: Yes

2. Has the statistical analysis been performed appropriately and rigorously? 

Reviewer #1: Yes

Reviewer #2: Yes

3. Have the authors made all data underlying the findings in their manuscript fully available?

Reviewer #1: Yes

Reviewer #2: Yes

4. Is the manuscript presented in an intelligible fashion and written in standard English?

Reviewer #1: No

Reviewer #2: Yes

5. Review Comments to the Author

Reviewer #1: Chen et al present a systematic review and meta-analysis on mortality and cardiovascular outcomes in dialysis patients treated with high vs. low dialyzable beta blockers. 4 retrospective studies were included. The authors found no difference in mortality, MACE, or MI but a significant difference in heart failure. While the study is overall clear and analytic methods seem overall appropriate, I do have several questions/comments for the authors to consider:

1. The English needs to be improved throughout the paper prior to being suitable for publication.

2. The study compares high vs. low dialyzable beta blockers. It would be helpful to inform the reader what threshold is used to determine high vs. low dialyzability.

3. I may have missed this but how was MACE defined? I assume acute MI and heart failure were included in this (though they are also listed as separate outcomes? Were any other cardiovascular outcomes included in MACE?

4. One of the included studies included peritoneal dialysis patients. I find it questionable of whether to include these patients given the inherent differences in dialysis modalities. For instance, given the much slower/prolonged clearance with peritoneal dialysis, one would think the contribution of intradialytic hypotension from beta-blocker clearance would be much less significant in this population.

5. I am confused by the heart failure results. The abstract states that "HDBBs were associated with significant reduction of risk in HF compared with individuals with LDBBs". However, when I look at Figure 2D which is showing heart failure outcomes seems to show that the results favor LDBBs which seems contrary to the prior statement. Further, this conclusion is drawn from only two studies, one (Wu) which contributes 97% of the relative weight while the other (Weir) is questionable as per the authors as bisoprolol may have been included in the wrong HDBB/LDBB category in that study. This draws into question any conclusions that can be drawn from this analysis.

Reviewer #2: The present is an interesting and timely meta-analysis

Some issues should be addressed

Abstract. Sentence in the results from "The risk of MACE to............]." appeared of no clear meaning and truncated

Abstract. It is not clear if the present results were in hospital or at which time of follow up

Methods. It is not clear if results derived from multivariable adjustement or not

Methods. Meta-regression for anamnesis of CAD, of HF and of EF (if available) should be added

Results. Quality of studies should be more expanded and subgroup analysis for it should be added

6. PLOS authors have the option to publish the peer review history of their article (what does this mean?). If published, this will include your full peer review and any attached files.

Reviewer #1: No

Reviewer #2: **Yes: **Fabrizio D'Ascenzo

---

## [Author Response · Author response to Decision Letter 0]

5 Dec 2022

To reviewer 1

#1. The English needs to be improved throughout the paper prior to being suitable for publication. Response:

Thanks for your suggestion. Our manuscript had been edited by experts in English.

#2. The study compares high vs. low dialyzable beta blockers. It would be helpful to inform the reader what threshold is used to determine high vs. low dialyzability.

Response:

Thank you for your comment. There was no definite cut-off value for determination of high dialyzable and low dialyzable beta blockers[1-4]. We had modified terms of high dialysable beta-blockers (HDBBs)/ low dialysable beta-blockers (LDBBs) to dialyzable beta-blockers (DBBs)/ non-dialyzable beta-blockers (NDBBs) in the modified version of manuscript. Moreover, the dialyzable thresholds of the beta-blockers were not mentioned in all included studies. According to Tieu’s study[1], only 10% of medication had reported definitive information about the threshold for drugs’ dialyzability. The classification of cardio-selective BBs and non-cardio-selective BBs in Shireman and Assimon’s studies[5, 6], it is also the categories by the dialyzability according to several studies about pharmacokinetics of beta-blockers[1, 2]. (DBBs, Cardio-selective BBs: Atenolol, metoprolol, nadolol; NDBB, Non-Cardio-selective: carvedilol, labetalol, and propranolol). Additionally, the classification of DBBs and NDBBs was based on studies of pharmacokinetics of beta-blockers[7-12] in Weir’s and Wu’s study[13, 14]. We had added the description for the detailed information in the introduction and result as following, Line 66” However, there was no definite threshold for differentiating between DBBs and NDBBs.”, Line 169” The classification of DBBs and NDBBs was based on the pharmacokinetics of beta-blockers (17-22) in two of the included studies (10, 12), while the classification of the other two studies was based on the cardio-selectivity of beta-blockers (13, 23). Previous studies have shown that cardio-selective and non-cardio-selective beta-blockers are pharmacologically equivalent to DBBs and NDBBs, respectively (11, 24).”

#3. I may have missed this but how was MACE defined? I assume acute MI and heart failure were included in this (though they are also listed as separate outcomes? Were any other cardiovascular outcomes included in MACE?

Response: 

Thank you for your comment. Several MACE definitions had been elucidated [15] including 3-point MACE, 4-point MACE and 5-point MACE. The classical 3-point MACE is composed of nonfatal stroke, nonfatal myocardial infarction, and cardiovascular death. The 4-point MACE is defined as classical 3-point MACE, with other events, like hospitalization for unstable angina or revascularization procedures. Finally, the definition of 5-point MACE is the result of 4-point MACE, and heart failure.[15] However, the MACE in the included studies were a little bit different, and it had been expressed at the response Table 1. Separately.

Response Table 1.

 MACE definition

Weir (2015) cardiovascular death, myocardial infarction, heart failure

Wu (2020) acute coronary syndrome, ischemic stroke and heart failure

Abbreviations: MACE, major adverse cardiovascular event

We had added the description for the detailed information in the result, line 117, as following, “The definition of MACE was based on that of each included study.” And added on” There was a slight variation in the definition of overall MACE between the two studies; while it involved cardiovascular death, myocardial infarction, and heart failure in one of the studies (12), the other referred it to acute coronary syndrome, ischemic stroke, and heart failure (10).” in line 231.

#4. One of the included studies included peritoneal dialysis patients. I find it questionable of whether to include these patients given the inherent differences in dialysis modalities. For instance, given the much slower/prolonged clearance with peritoneal dialysis, one would think the contribution of intradialytic hypotension from beta-blocker clearance would be much less significant in this population.

Response:

Thanks for your comment. Theoretically, the blood concentration of a drug would be steadier in peritoneal dialysis patient than in the hemodialysis patients due to the much slower/prolonged clearance of peritoneal dialysis. However, there are less studies discussing about the relation between drug dialyzability and peritoneal dialysis. According to Hirata’s study,[16] drug clearance for a given duration is markedly lower during CAPD than hemodialysis. Nevertheless, CAPD is a continuous procedure. Therefore, weekly clearance is not markedly different between CAPD patients and hemodialysis patients. On the other hand, we had provided sensitivity analysis which exclude the study composed of peritoneal dialysis population (S2 Table. Sensitivity analysis), and displayed no significant difference after sensitivity analysis.

#5. I am confused by the heart failure results. The abstract states that "HDBBs were associated with significant reduction of risk in HF compared with individuals with LDBBs". However, when I look at Fig 2D which is showing heart failure outcomes seems to show that the results favor LDBBs which seems contrary to the prior statement. Further, this conclusion is drawn from only two studies, one (Wu) which contributes 97% of the relative weight while the other (Weir) is questionable as per the authors as bisoprolol may have been included in the wrong HDBB/LDBB category in that study. This draws into question any conclusions that can be drawn from this analysis.

Response:

Thanks for your comment. We had modified the Fig 2. Also, we had added on reviewer’s comments at discussion in line 370 as following,” However, this conclusion was drawn from only two studies, one (10) of which contributed to 97% of the relative weight while the finding of the other (12) was questionable because the authors classified bisoprolol into the DBB category.”

To reviewer 2

#1. Abstract. Sentence in the results from "The risk of MACE to............]." appeared of no clear meaning and truncated

Response:

Thanks for your comment. We had revised as the following (line 30)” Analysis of four observational studies including 75,193 individuals undergoing dialysis in hospital and community settings after a follow-up from 180 days to six years showed an overall all-cause mortality rate of 11.56% (DBBs and NDBBs: 12.32% and 10.7%, respectively) without significant differences in risks of mortality between the two groups [random effect, aHR 0.91 (95% CI, 0.81–1.02), p=0.11], overall MACE [OR 1.03 (95% CI, 0.78–1.38), p=0.82], and AMI [OR 1.02 (95% CI, 0.94–1.1), p=0.66].”

#2. Abstract. It is not clear if the present results were in hospital or at which time of follow up

Response:

Thanks for your comment. We had added on detailed description at line 30 as following” Analysis of four observational studies including 75,193 individuals undergoing dialysis in hospital and community settings after a follow-up from 180 days to six years” 

#3. Methods. It is not clear if results derived from multivariable adjustement or not

Response:

Thanks for your comment. We had revised the description in method (line 140) as” The odds ratio (OR) and 95 % confidence interval (CI) were extracted from the included studies. In addition, the hazard ratio (HR) of mortality was also analyzed. For studies that provided information on HR, the data were used for analysis.” We not only expressed the pooled result with aHR (Fig 2A) [aHR: 0.91; 95% CI, 0.81–1.02, p = 0.11], but also demonstrated a consistent result for mortality with OR (S6 appendix).

#3. Methods. Meta-regression for anamnesis of CAD, of HF and of EF (if available) should be added

Response:

Thanks for your comment. We had added meta-regression for anamnesis of CAD and of HF. (S3 Table 3) However, there was no EF data available. We had added meta-regression in result as “The quantitative measures of CAD and HF were not associated with all-cause mortality. (CAD, Z = 1.65, p = 0.10; HF, Z = 1.28, p = 0.20, sTable 3)” (line 285)

#4. Results. Quality of studies should be more expanded and subgroup analysis for it should be added

Response:

Thanks for your comment. We had added subgroup analysis related to the mortality rates based on scores of Newcastle-Ottawa Scale Quality Assessment for included studies (NOS score =8 versus NOS score =9). (line 266-272) We revised the manuscript as following (line 266-272), “ In respect of the impact of the quality of the included studies, analysis of the study with a NOS of eight showed a higher but non-significant increase in mortality risk in the DBB group compared to that in the NDBB group [Fig 3D, random effect, OR = 1.37 (95% CI, 0.93–2.0), p = 0.12]. In contrast, investigating the other three studies reporting a NOS score of nine demonstrated a non-significant reduction in mortality risk among those receiving DBBs compared with those taking NDBBs [Fig 3D, random effect, OR = 0.85 (95% CI, 0.71–1.02), p = 0.09].”

---

## [Decision Letter · Decision Letter 1]

12 Dec 2022

Impact of type of dialyzable beta-blockers on subsequent risk of mortality in patients receiving dialysis: A systematic review and meta-analysis

PONE-D-22-24265R1

Dear Dr. Chen,

We’re pleased to inform you that your manuscript has been judged scientifically suitable for publication and will be formally accepted for publication once it meets all outstanding technical requirements.

Kind regards,

Gianpaolo Reboldi, MD, MSc, PhD

Academic Editor

PLOS ONE

Additional Editor Comments (optional):

Reviewers' comments:

Reviewer's Responses to Questions

**Comments to the Author**

1. If the authors have adequately addressed your comments raised in a previous round of review and you feel that this manuscript is now acceptable for publication, you may indicate that here to bypass the “Comments to the Author” section, enter your conflict of interest statement in the “Confidential to Editor” section, and submit your "Accept" recommendation.

Reviewer #1: All comments have been addressed

Reviewer #2: All comments have been addressed

2. Is the manuscript technically sound, and do the data support the conclusions?

Reviewer #1: Yes

Reviewer #2: (No Response)

3. Has the statistical analysis been performed appropriately and rigorously? 

Reviewer #1: Yes

Reviewer #2: (No Response)

4. Have the authors made all data underlying the findings in their manuscript fully available?

Reviewer #1: Yes

Reviewer #2: (No Response)

5. Is the manuscript presented in an intelligible fashion and written in standard English?

Reviewer #1: Yes

Reviewer #2: (No Response)

6. Review Comments to the Author

Reviewer #1: The authors have adequately responded to all prior comments.

No concerns regarding dual publication, research ethics, or publication ethics.

Reviewer #2: (No Response)

7. PLOS authors have the option to publish the peer review history of their article (what does this mean?). If published, this will include your full peer review and any attached files.

Reviewer #1: No

Reviewer #2: **Yes: **Fabrizio D'Ascenzo

---

## [Editor Report · Acceptance letter]

22 Dec 2022

PONE-D-22-24265R1 

Impact of type of dialyzable beta-blockers on subsequent risk of mortality in patients receiving dialysis: A systematic review and meta-analysis 

Dear Dr. Chen:

I'm pleased to inform you that your manuscript has been deemed suitable for publication in PLOS ONE. Congratulations! Your manuscript is now with our production department. 

Kind regards, 

on behalf of

Prof Gianpaolo Reboldi 

Academic Editor

PLOS ONE